# Epigenetics, Enhancer Function and 3D Chromatin Organization in Reprogramming to Pluripotency

**DOI:** 10.3390/cells11091404

**Published:** 2022-04-21

**Authors:** Andreas Hörnblad, Silvia Remeseiro

**Affiliations:** 1Umeå Centre for Molecular Medicine (UCMM), Umeå University, 901 87 Umeå, Sweden; 2Wallenberg Centre for Molecular Medicine (WCMM), Umeå University, 901 87 Umeå, Sweden

**Keywords:** epigenetics, enhancer, 3D genome, reprogramming, pluripotency, iPSCs, OSKM

## Abstract

Genome architecture, epigenetics and enhancer function control the fate and identity of cells. Reprogramming to induced pluripotent stem cells (iPSCs) changes the transcriptional profile and chromatin landscape of the starting somatic cell to that of the pluripotent cell in a stepwise manner. Changes in the regulatory networks are tightly regulated during normal embryonic development to determine cell fate, and similarly need to function in cell fate control during reprogramming. Switching off the somatic program and turning on the pluripotent program involves a dynamic reorganization of the epigenetic landscape, enhancer function, chromatin accessibility and 3D chromatin topology. Within this context, we will review here the current knowledge on the processes that control the establishment and maintenance of pluripotency during somatic cell reprogramming.

## 1. Introduction

The development of multicellular organisms occurs through a series of cell divisions, morphogenetic processes and the ability of cells to adopt different cell fates. From one single totipotent cell, the zygote, a succession of complex events and intermediate cell states gives rise to all organs and tissues, composed of hundreds of different cell types, that constitute the entire individual. The variety of cell types, all sharing their unique genetic information, arises as the differential usage of the genomic sequence is induced in response to environmental cues, which translates into the activation of distinct gene regulatory networks that determine gene expression programs characteristic of each cell type. These cell-type-specific gene-expression programs are established and maintained via crosstalk between transcription factors (TFs) and the chromatin environment [1]. Transcription factors bind to specific DNA sequence motifs located in regulatory elements (i.e., gene promoters and enhancers) [2] and can act as activators or repressors. This function depends on whether they favour or block the recruitment of molecular complexes involved in transcription, DNA or histone modification, or chromatin remodelling. Some TFs have dual functions and act either as activators or repressors depending on the nuclear environment [3,4,5]. Chromatin modifications affect the accessibility to DNA and, therefore, promote or impede the binding of transcription factors to their regulatory regions. Some TFs act as “pioneer transcription factors” [6] given their ability to also bind inaccessible regions and initiate chromatin remodelling required for transcriptional regulation.

In the linear genome regulatory regions, such enhancers can be located very far away from the target promoters, which are adjacent to the transcription start sites (TSSs). The long-range action of enhancers over gene promoters becomes possible given the structural organization of the genome in 3D space. This topological organization of the genome allows for physical proximity between enhancers and their target promoters, while limiting or preventing undesired interactions which would cause ectopic expression. This 3D organization sets the stage for the specificity of enhancer-promoter interactions, and thus constitutes a functional partition of the genome [7]. Transcription factors must therefore exercise their regulatory functions in the context of the 3D-chromatin organization. The interplay between the regulatory and architectural organization of the genome results in the expression of particular and fine-tuned transcriptional programs during differentiation into specific cell-types.

Key to embryonic development of multicellular organisms are the pluripotent cells that have the potential to differentiate into cells of all three primary germ layers. These cells share their self-renewal capacity with adult stem cells that ensure the maintenance of tissue homeostasis during postnatal life. A long-standing question has been how cells achieve and maintain this pluripotency. It was long believed that differentiated cells were irreversibly committed to a cell-fate and could not return to an earlier progenitor or pluripotent state. However, pioneering experiments of cell nucleus transfer [8] and, more recently, induction of pluripotency via ectopic expression of transcription factors [9,10] or even by treatment with small-molecule compounds [11], have demonstrated that differentiated cells can indeed be reprogrammed into a pluripotent state, or even directly to other cell types (reviewed in Wang et al. 2021 [12]). As such, these experimental models have not only demonstrated the extensive plasticity of differentiated cells, but also provided important contributions in the study of mechanisms underlying cell differentiation and specification. The reprogramming of fibroblasts into induced pluripotent stem cells (iPSCs) by ectopic expression of Oct4, Sox2, Klf4 and cMyc (jointly known as OSKM), constitutes a fundamental discovery by Takahashi and Yamanaka [9], which contributed to the understanding of cellular identities and opened-up the multiple practical applications of iPSCs. Transcription-factor-induced reprogramming changes the transcriptional profile and chromatin landscape of the starting somatic cell to the resulting pluripotent cell, which functionally mimics an embryonic stem cell (ESC) [9]. This function of the reprogramming factors is exerted by their capacity to bind closed chromatin and induce chromatin changes in the early stages of reprogramming, prior to the major transcriptional changes that take place during the process, in which the somatic program must be switched-off and the pluripotent program established.

In the last decades, the fast development of next generation sequencing technologies has greatly advanced our knowledge on processes underlying gene expression control and induction of gene regulatory networks. Different approaches now allow us to obtain transcriptome profiles (RNA-seq, scRNA-seq), determine genome-wide transcription factor binding (ChIP-seq, Cut&Run [13,14]) and chromatin accessibility (DNase-seq and ATAC-seq [15,16]), and study higher order chromatin conformation with 3C-based technologies [17,18,19,20,21,22,23]. Integration of data obtained by mapping different chromatin features genome-wide has furthered our understanding of the epigenetic changes that precede or accompany cell-specific gene expression programs and allowed us to start identifying non-coding regulatory elements in the genome (e.g., enhancers) that control cell-type specific gene expression. The theorical frameworks developed by diverse computational studies [24,25] and mathematical modelling [26,27,28,29,30] have also contributed to our comprehension of how key changes in regulatory networks determine cell fate transitions during embryonic development and reprogramming to pluripotency. In addition, the advent of highly efficient genome and epigenome editing techniques (e.g., CRISPR/Cas9, CRISPRi/a) has initiated the functional interrogation of such elements with a higher throughput [31,32,33,34,35,36,37], thus shedding light on their in vivo contribution to gene expression programs.

Here, we review the relation between epigenetics, 3D chromatin organization and the remodelling of enhancer activity during somatic cell reprogramming to pluripotency.

## 2. Epigenetics of Reprogramming

Reprogramming of somatic cells is driven by the activity of ectopically expressed transcription factors, which bind to target sequences in the genome and induce transcriptional changes, epigenetic remodelling and reorganization of the 3D chromatin architecture.

The simplicity and robustness of the OSKM-mediated induction of iPSCs has allowed for reprogramming of multiple different somatic cells to pluripotency. This reprogramming process is accompanied by changes in the transcriptional profile and chromatin states, from that of the somatic cell to the transcriptome and chromatin state of a pluripotent cell and involves the transition through different intermediate states. The importance of OSK for the pluripotent state resides in the fact that Oct4, Sox2 and Klf4 tend to colocalize at many cell-type-specific enhancers in embryonic stem cells (ESCs), together with additional pluripotency transcription factors (e.g., Nanog, Klf2, Sall4, etc.) [38,39]. Enhancers are crucial regulatory elements for the establishment and maintenance of cell-type-specific gene expression programs [40]. In this context, enhancers integrate multiple pluripotency transcription factors and signalling cues which results in the expression of various pluripotency genes and a stable ESC gene expression program. Although part of the original reprogramming cocktail, cMyc is distinct from the OSK reprogramming factors in that it is not a component of the core pluripotency network [38,39]; it predominantly associates to promoters and it is not necessary for the reprogramming process [41,42]. cMyc acts instead as an amplifier of transcriptional outputs arising due to OSK action, likely by binding the core promoter of active genes that are initially expressed at low levels, but also by rewiring biosynthetic pathways required for the initiation of reprogramming [43]. Amplifying such expression changes, cMyc would potentially help to trap genes in the ON state by predominantly binding to core promoters, since it is typically not present at enhancers [38,44,45].

Comparison of OSKM-occupancy in the early reprogramming state *versus* the pluripotent state showed that 85% of the initial binding events occur in regions distal to gene promoters, including some known enhancers [45]. Therefore, in early reprogramming, occupancy at promoters and transcriptional activation of many target genes is preceded by binding of the reprogramming factors to distal elements. This appears to be important for subsequent chromatin remodelling and epigenetic changes that lead to activation of the pluripotency expression program.

### 2.1. Pioneer Factors in Reprogramming

During development, pioneer transcription factors initiate a cascade of events that eventually lead to activation of lineage-specific genes. In early developmental stages, pioneer factors access tissue-specific enhancers, induce chromatin de-condensation and chromatin remodelling; and prime enhancers and promoters for binding of additional transcription factors at later stages, which leads to transcriptional activation of lineage-specific transcriptional programs.

Reprogramming factors are recruited to DNA through their target sequence motif and act in a concentration-dependent manner. Given that reprogramming factors are ectopically expressed at high levels, it may be important for reprogramming that the nuclear concentration of OSKM is sufficiently high to also occupy sites of lower affinity [45]. In addition, transcription factor binding is dependent on a permissive chromatin environment, which favours the accessibility to their DNA sequence motifs and therefore the capability of transcription factors to engage their targets (Figure 1). Reprogramming factors do bind to open chromatin marked by active histone modifications, among which cMyc binding is more strongly associated to pre-existing active chromatin than OSK [45,46]. However, the reprogramming factors also have the capacity to bind closed chromatin and drive chromatin changes early in reprogramming (Figure 1), before the occurrence of major transcriptional changes. In early reprogramming of human fibroblasts, around 70% of the reprogramming factor binding sites are located within closed chromatin regions [45]. In particular, this applies to OSK, while the ability of cMyc to access closed chromatin is limited and depends on the preceding occupancy of the former three. cMyc then potentiates the action of OSK binding rather than initiating the binding events itself; whereas each of the OSK factors can target closed chromatin sites alone, without the other two factors located at such sites [45]. Pioneer transcription factors [47] can induce such an epigenetic priming, paving the way for subsequent recruitment of other factors, eventually leading to transcriptional activation of a particular regulatory network. In reprogramming, Oct4, Sox2 and Klf4 would act as pioneer factors binding closed chromatin on their own, as proposed by Zaret and colleagues and supported by additional evidence [48,49,50,51], eventually leading to activation of the pluripotency program. Additional studies support the contribution of not only individual reprogramming factors, but also combinations of factors [52,53,54]. Noteworthy is the cooperative binding of transcription factors during reprogramming. In early reprogramming, pluripotency enhancer selection occurs in a stepwise manner through the collaborative binding of OSK at sites of high OSK-motif density [55]. Stage-specific TFs influence OSK occupancy, while cooperativity among OSK and with stage-specific TFs directs somatic-enhancer inactivation and pluripotency-enhancer selection to boost reprogramming to pluripotency [55].

In the context of reprogramming, the fact that OSK can act as pioneer factors mimics the scenario of developmental decisions, where pioneer factors occupy enhancers in early stages. The pioneer factor properties of the reprogramming factors are pivotal in subsequent steps of iPSC generation as the rewiring of the transcriptional program is accompanied by both nucleosomal reorganization and extensive epigenetic modifications [56].

### 2.2. Early Reprogramming Changes in Enhancers and Promoters

The abrogation of the somatic gene expression program constitutes the first step in somatic reprogramming. The initial expression changes are limited to genes that are in an open and accessible chromatin state, i.e., initial binding of pluripotency factors occurs largely in promoters bound by active chromatin marks in the fibroblasts, and it correlates with both up- and down-regulation of the corresponding genes [57]. Concomitantly, extensive chromatin remodelling, such as changes in histone modifications, occurs genome-wide immediately after expression of the reprogramming factors (Figure 1). In the case of H3K4me2 (associated to active or poised promoters and enhancers), the gain of binding to many promoter regions largely correlates with the absence of transcriptional changes in the initial stages and transcriptional activation in later stages during reprogramming. In fact, in early reprogramming, remodelling of histone marks is more extensive than gene expression changes [57]. Moreover, the initial transcriptional changes take place in a large fraction of the cells, even though completion of reprogramming only occurs in less than 1% of the starting cell population [58], suggesting that the major hurdle in the reprogramming trajectory is to achieve transcriptional activation of the more inaccessible regions that require chromatin remodelling.

Enhancers are *cis*-regulatory elements which contribute to the specific expression pattern of a certain target gene, both temporally and spatially. Active enhancers are highly accessible chromatin regions decorated by active histone modifications (H3K27ac, H3K4me1, H3K4me2) [59,60], marked by chromatin regulators p300 and BRG1 [60], and bound by RNA Polymerase II (RNA PolII). In their inactive state, enhancers are defined as either silenced, primed or poised, depending on their combination of histone marks and other chromatin features [60,61,62,63]. Like active enhancers, poised enhancers can be distally bound by p300 and H3K4me1; however, they are distinguished by the absence of H3K27ac and enrichment of H3K27me3 (also, represented as H3K4me1+, H3K27ac−, H3K27me3+) [60,64]. H3K4me1+, H3K27ac− and H3K27me3− are marks of primed enhancers [65], while the binding of H3K27me3 in the absence of other marks represents the polycomb-repressed silenced state [59]. Thus, the chromatin state of *cis*-regulatory enhancers correlates with their regulatory activity and is important for their function. Studies have shown that complex regulatory landscapes are frequently present in developmental gene loci involved in embryonic patterning, cell specification and differentiation; and appear also to be a prominent feature in the regulatory network of pluripotent stem cells [66]. In particular, it was suggested that pluripotency genes in embryonic stem cells are controlled by large regulatory regions (i.e., super-enhancers) containing multiple transcriptional enhancers characterized by high occupancy of master transcription factors as well as the mediator complex [66].

In the context of pluripotency, and in contrast to promoters, changes in chromatin at enhancers are more remarkable in the early phases of reprogramming [57]. Around 85% of the OSKM binding sites at this early stage of reprogramming are located at distal regions, including known regulatory regions, rather than gene promoters [45]. A major fraction of somatic enhancers switch state before the first cell division, while many pluripotency specific enhancers are becoming established [57], as evidenced by changes in H3K4me1/me2. Given that H3K4me1/2 marks both active and poised enhancers, one cannot distinguish between these two states, but it seems likely that gain of this histone modification in ESC-specific enhancers marks a poised state. The acquisition of poised ESC-specific enhancers early in reprogramming would allow a quick switch to the active enhancer state at later stages of reprogramming when it would be crucial to efficiently coordinate the activation and expression of ESC-specific genes. This is also consistent with cell fate specification in the context of development, where transcriptional activation of lineage-specific genes is preceded by local chromatin remodelling and epigenetic priming in their enhancer(s) and promoter regions [60,62,63,67,68]. Altogether, these chromatin remodelling events (Figure 1) are consistent with a switch-off of the somatic transcriptional program as a major early event during reprogramming, and that this precedes the complete switch-on of the pluripotency transcriptional program.

### 2.3. Epigenetic Barriers to Reprogramming

Pioneering binding of OSK to closed chromatin regions in early stages is a critical step in reprogramming, but it does not occur at all regulatory regions that are active in the pluripotent cells. In fact, many regulatory regions occupied by OSMK in the pluripotent state are not initially bound by these factors [45,46]. Thus, the generation of a favourable chromatin landscape that allows for re-engagement of the pluripotency transcriptional program appears to be one of the main barriers in the process, which is also illustrated by the much lower frequency of cells that reach this step in the reprogramming trajectory [58]. Two of the features that underlie this difficulty are DNA methylation and the heterochromatin mark H3K9me3 (Figure 1). Both negatively influence the binding of the reprogramming factors and, as such, contribute to maintain a somatic cell identity.

DNA methylation restricts early reprogramming events [37,52,53] and seems to provide a barrier that restricts changes in histone modifications in a rate-limiting step and prevents the recruitment of reprogramming factors [51]. Pluripotency-specific promoters with high levels of DNA methylation in the starting fibroblasts convert to a demethylated state late in reprogramming, concomitantly to the gain of active chromatin marks at these locations [57]. The gain of active chromatin modifications at enhancers and promoters early in reprogramming correlates with hypomethylated states during the entire process. The cells that fail to demethylate genomic regions important for the pluripotency transcriptional program at later stages fail to complete reprogramming [69]. TET proteins play an important role in de-methylation; it has been shown that TET2 physically interacts with Klf4, C/EBPα, Tfcp2l1 transcription factors, as well as having a major fraction of overlapping DNA binding sites with Klf4 in naïve pluripotent cells [70]. ESRRB is another transcription factor that has been shown to drive de-methylation and nucleosome displacement, eventually allowing for recruitment of reprogramming factors [71]. Thus, it appears that de-methylation occurs in a stepwise manner via TF-guided recruitment of TET proteins, and that this precedes opening of the chromatin at these loci [53,70,71,72]. In addition, downregulation of Dnmt1 (DNA methyltransferase 1), which methylates preferentially hemi-methylated sites, favours reprogramming to pluripotency [69], further supporting that DNA methylation limits the reprogramming process. Nonetheless, the de novo methyltransferases Dnmt3a/b are dispensable for reprogramming [73]. While the changes in histone modifications and gene expression seem to occur gradually throughout the process, the methylation of somatic genes and the demethylation of some pluripotency genes take place late in reprogramming.

On the other hand, the heterochromatin mark H3K9me3 also constitutes a hurdle to reprogramming as it partly limits the access of the reprogramming factors. Inhibition of SUV39H1, the methyltransferase responsible for H3K9 methylation, increases the efficiency of reprogramming and the binding of OSKM [45]. H3K9me3 maintenance is stimulated by TGFβ signalling, which upon inhibition decreases the signal of H3K9me3 regions and improves reprogramming efficiency [45,74]. This may underlie the fact that an important hallmark of early reprogramming is the mesenchymal to epithelial transition (MET) that includes downregulation of typical mesenchymal genes such as Snai1, Snai2, Zeb1 and Zeb2, and of which TGFβ is a major inducer [75,76].

The difficulty of overcoming these epigenetic barriers in reprogramming is evidenced by the retention of an “epigenetic memory”, i.e., iPSCs maintain residual histone and DNA methylation signatures characteristic of the somatic cell of origin [77,78], which distinguish them from ESCs. Retention of such epigenetic memory influences the in vitro differentiation potentials of iPSCs, since it favours the differentiation into lineages related to the donor cell and restricts transition into other less-related cell fates.

Apart from the importance of DNA de-methylation and the removal of heterochromatin histone marks, enhancer-mediated initiation of transcription requires further reorganization and modulation of the chromatin environment. Once active, enhancers are highly accessible regions decorated with active histone modifications (e.g., H3K27ac) and PolII binding. These chromatin marks are not acquired simultaneously, and it has been shown, in the context of development and reprogramming, that activation of enhancers is preceded by priming via the action of pioneer transcription factors [60,62,63,79,80,81].

Even so, functional interactions between active enhancers and their target genes require close physical proximity in the 3D genome. Thus, rewiring of transcriptional programs is not only associated with major remodelling of DNA methylation, histone modifications and chromatin accessibility, but also accompanied by extensive reorganization of the 3D genome topology (i.e., the folding of the genome). In the following section, the reorganization of the 3D genome during reprogramming is discussed.

## 3. 3D-Chromatin Organization in Pluripotency

The higher-order organization of the genome is intimately associated with gene regulation and subsequently to the establishment of different cell identities, not only during development, but also during the reprogramming process. The activation of transcription factors, in response to environmental cues, drives the establishment of cell-type specific transcriptional programs and therefore determines cell-fate decisions. Given that chromatin modifications modulate the access of transcription factors to regulatory elements, it is such interaction between transcription factors and the chromatin landscape that governs the establishment and maintenance of gene expression programs characteristic to each cell type [1]. In addition to histone modifications and DNA methylation (discussed in the previous section), the 3D organization of the genome constitutes an important part of the chromatin landscape. The fact that the genome presents a topological organization has functional implications in diverse nuclear processes, including transcription [82,83], and provides a scaffold for functional contacts between distant enhancers and their target gene promoters. This functional interplay between the topological and the regulatory organization of the genome is a central question, not only in cell fate decisions, but also in pluripotency and reprogramming.

### 3.1. Spatial Organization of the Genome and Gene Regulation

An essential part of gene expression regulation is achieved at the transcriptional level, mostly through the integrated action of *cis*-regulatory elements, including core promoters, as well as distal regulatory elements such as enhancers, silencers, insulators and tethering elements. Cell-type specific transcriptional programs are established through the interplay between proteins that control transcriptional activity and their interaction with such regulatory sequences of the genome.

In particular, *cis*-acting enhancer elements are important for the initiation of gene expression programs characteristic of each cell type. Enhancers can be located very far away from their target promoters in the linear genome, and often several enhancers with overlapping activity control cell-type specific gene expression [84,85,86,87,88,89], also reflecting the complexity of gene expression patterns. Importantly, enhancers require physical proximity with the promoter(s) of their target genes to activate gene expression. Thus, the dynamic folding of the genome is intrinsically connected to the control of gene expression, and it dictates the structural framework in which enhancers perform their function.

In the last decades, our understanding of the interplay among 3D-chromatin organization, histone modifications and other epigenetic marks, and transcriptional control has significantly increased. It is evident that these features have important functions for the specification and maintenance of cellular identity. Improvements in microscopy, as well as chromosome-conformation-capture-based sequencing technology, has revealed a highly intricate spatial organization of the genome at multiple levels. In the nucleus, individual chromosomes are segregated into distinct territories (Figure 2a). The gene-rich transcriptionally active regions within each chromosome tend to be localized towards the interior of the nucleus, defined as A compartments, while transcriptionally inactive gene-poor regions occupy B compartments that to a large extent also overlap with lamina-associated domains (LADs) in the periphery of the nucleus [20,90]. The A and B compartments can then further be divided into topologically associating domains (TADs) [91,92] that are regions defined at (sub)megabase scale based on their self-interacting properties (Figure 2a). It is within these self-interacting TADs that cell type specific chromatin contacts between distant genomic regions occur, in the range of Kb to Mb away in the linear genome. Among these long-range contacts, we find interactions between regulatory elements (e.g., enhancer-promoter), repressive/poised interactions and cohesin/CTCF-bound regions. TADs have been shown to play an important role in gene regulation by constraining the range of action of enhancers in the linear genome [93,94], as illustrated by the high overlap between TADs and regulatory domains [95]. This constraint decreases the probability of enhancers to act on non-target genes and increases the specificity of enhancers by facilitating interaction with their target promoters [93,94,96].

TADs appear to be largely cell type invariant and may thus provide a structural scaffold onto which functional units can be organized, depending on the cellular context, since boundaries of regulatory interactions, chromatin states and transcriptional activity correlate well with TAD transitions [21,92,95]. Even if TADs are largely conserved, both the strength of insulation of TAD boundaries and the intra-TAD interactions can differ significantly across cell types [97]. Although transition from one cellular identity or state to another does not appear to induce major relocations of TAD boundaries, compartment switching and changes in intra-TAD structure are both inherent to the rewiring of transcriptional programs and cell-type specification (Figure 2b). In addition, during cell-fate transitions, activating and repressive/poised interactions can be reorganized [98].

The different levels of 3D genome organization are established due to two phenomena: loop extrusion and compartmental segregation. In the loop extrusion [99,100] structural proteins impose chromatin looping, i.e., the ring-shaped cohesin complex slides through and extrudes DNA until it is haltered by convergently oriented CTCF sites. Cohesin/CTCF-mediated loops, TADs, sub-TADs [101] and “stripes” [100,102,103,104] are formed by loop extrusion. Of note is the specialized contribution of the different cohesin-SA1 and cohesin-SA2 complexes to 3D genome architecture [105], also in the context of embryonic stem cells [106]. On the other hand, the A and B compartments are formed by self-aggregation of chromatin with similar features, such as transcriptional activity and chromatin modifications, and are considered to be driven, at least in part, by phase separation mechanisms (i.e., molecules separate into discrete liquid condensates in the absence of membranes) [107]. Self-aggregating properties giving rise to nuclear condensates have been described for many molecules involved in transcriptional regulation, including transcription factors [108,109], chromatin modifiers and members of the transcriptional machinery [110,111], and also fit well with data describing transcription factories [112] or 3D enhancer hubs [113,114,115,116], among others. Although these processes may not explain all aspects of the spatial organization of the genome, the interplay between these two underlying mechanisms is an important component in shaping the regulatory genome and the cell-type specific regulatory programs, including pluripotency.

### 3.2. 3D-Chromatin Reorganization during Reprogramming to Pluripotency

Concurrent with epigenetic priming and the remodelling of chromatin marks, the spatial organization of the genome is restructured during the reprogramming process. Pluripotent stem cells, including both ESCs and iPSCs, present an epigenetic environment characterized by “open” and permissive chromatin (i.e., low levels of DNA methylation and heterochromatin marks, and the presence of H3K27me3-repressed regions and bivalent domains at lineage-specification genes) [117]. This epigenetic landscape changes during cell differentiation and needs to be re-established upon reprogramming. Regarding the architectural organization, iPSCs generated from different somatic cell types display a 3D genome topology very similar to embryonic stem cells (ESCs) [118]. Comparison of the starting somatic cells with the resultant iPSCs has shown that 3D chromatin organization is dramatically restructured during reprogramming, not only at specific loci [119,120], but also genome-wide [118].

3D chromatin organization is highly connected to cell identity. The genome-wide 3D organization of iPSCs, resulting from reprogramming of various somatic cells, has shown that, at the compartment level, iPSCs are virtually identical to ESCs [118]. Thus, the reprogramming process ensures an accurate reorganization back to the topological organization typical of the pluripotency state. Notably, this is a dynamic process with a particular kinetics characterized by the presence of transient and intermediate topologies, given that changes in the compartmentalization occur gradually along the reprogramming process and are associated with transcriptional changes [121]. In addition, epigenetic remodelling accompanies or precedes the compartmental changes in the mentioned study performed in B-cell reprogramming, where the vast majority of TAD boundaries also remained unaltered during the reprogramming process [121]. Even though the changes in TADs were minimal, a big fraction of TADs underwent alterations in the insulation strength of the boundaries or the frequency of intra-TAD interactions, a phenomenon associated with transcriptional changes, and likely promoting or preventing communication between enhancers and promoters (Figure 2b).

Besides the large-scale 3D chromatin reorganization, long-range chromatin loops also suffer a dynamic rewiring during reprogramming, for instance, involving chromatin interactions at key stem cell genes including Nanog or Oct4 [119,122,123]. A study mapping the enhancer-promoter interactome by H3K27ac-HiChIP determined multiple cell-type specific regulatory interactions, and it further supported the dramatic enhancer rewiring that takes place during reprogramming in association with the transcriptional status of the linked gene promoter [22,124].

Recent studies have elucidated the presence of highly interacting enhancers, named 3D enhancer hubs [125]. Multiple genes that are activated in a coordinated manner during reprogramming interact with PSC-specific enhancer hubs. Thus, 3D enhancer hubs would constitute important architectural pillars of cell identity; and, being reorganized during reprogramming and cell fate decisions, they would play a key role in gene coregulation at large-scale. However, many questions require to be addressed with regards to how 3D enhancer hubs assemble and how they operate.

### 3.3. Transcription Factors as Drivers of 3D Chromatin Organization

Somatic cell reprograming is driven by the ectopic expression of the OSKM reprogramming factors, which, upon binding to specific genomic regions in the somatic cells, can impose a series of molecular events that range from epigenetic remodelling to transcriptional changes along the reprogramming process. Recent studies have investigated the role of transcription factors (TFs) in topological reorganization and hub formation in cell-fate transitions.

Early studies had already identified the named “transcription factories” as nuclear foci where the basal transcriptional machinery localizes and nascent transcription occurs [112]. Active genes and regulatory elements come together at transcription factories, which likely represent the currently termed “chromatin 3D hubs”, which contain at least two different transcription units at certain given times (promoter-promoter, promoter-enhancer, enhancer-enhancer) [126]. Functional studies have shown that depletion of transcription factors or modification of the TF binding sites affects enhancer-promoter interactions and gene expression [113,125,127,128,129,130]. In particular, disruption of Klf4 binding sites, as well as depletion of Klf4, Klf2 and Klf5, have shown that KLF factors play a critical role in the maintenance of the 3D enhancer network in pluripotent stem cells (PSCs) [125]. Nanog [131] and Sox2 [129] are additional TFs reported to mediate enhancer-promoter interactions in PSCs.

It has been suggested that TFs can mediate enhancer-promoter interactions in multiple ways, such as the phase separation model [108], through protein homo-dimerization [114] or oligomerization [130,132], via loop extrusion by interaction with cohesin/CTCF [131,133,134,135,136], or by recruitment of chromatin remodelers [137]. During reprogramming, the binding of OSKM can induce changes in the organization of TADs (e.g., the appearance of new/stronger boundaries [121]) or creation of new contacts [125]. Further studies are needed to understand through which mechanisms OSKM factors can induce local or genome-wide 3D chromatin reorganization during reprogramming.

### 3.4. Chromatin Accessibility during Reprogramming

Reprogramming of somatic cells to pluripotent cells involves the switch-off of the somatic program and establishment of the pluripotent state. Together with the epigenetic modifications and the topological changes, the dynamics of chromatin accessibility are also altered during reprogramming. In this context, the Yamanaka factor-induced reprogramming and ATAC-seq have been valuable tools to study chromatin accessibility during cell fate transitions. ATAC-seq (Assay for Transposase-Accessible Chromatin using sequencing) allows to capture open chromatin sites and infer of transcription factor occupancy genome-wide [15]. The analysis of chromatin accessibility by ATAC-seq in reprogramming has shown that the open somatic enhancers get closed at early stages of reprogramming, while the closed pluripotency-related enhancers are gradually opened [53]. Transcription factor motif search within these regions revealed that pluripotency associated TFs appeared only in the closed-to-open sites at early stages, including Oct, Sox and Klf, while other pluripotency factors (i.e., Tcf, Tfcp2L1 or Esrrb) only appeared at later stages. On the other hand, the open-to-closed sites displayed enrichment in motifs of somatic TFs, such as AP-1, TEAD and the RUNX family. This altogether supports the view that OSK reprogramming factors are not responsible for the closing of the somatic enhancers [53]. Additional studies using ATAC-seq have pointed to a differential role of the reprogramming factors Sox2 and Oct4 in chromatin accessibility. Oct4 and Sox2 seem to regulate chromatin accessibility at mostly distinct loci [138]. In addition, a stronger increase in accessibility is detected at sites bound by Sox2-only compared to sites bound by Oct4-only, though the most profound is observed at Sox2/Oct4-common sites [54]. Sox2 facilitates chromatin accessibility changes during reprogramming to a higher extent than Oct4, though Oct4 can boost chromatin opening when acting together with Sox2 [54]. Moreover, Oct4 is required through the cell cycle to maintain enhancer accessibility and to re-establish it at certain genomic positions during the mitosis to G1-phase transition [138]. Thus, chromatin accessibility dynamics are observed during somatic reprogramming and, concomitantly, throughout cell cycle progression during the reprogramming process.

## 4. Perspectives

Ectopic expression of reprogramming factors has proven to be a very valuable tool for the study of different aspects of cell-fate conversions. Here, we give an overview of the major events occurring in reprogramming to pluripotency in terms of epigenetic changes, enhancer function and 3D chromatin reorganization.

The rapid technological development in the last decades has brought about a huge leap in the understanding of gene regulation and how epigenetics, chromatin modifications and 3D genome organization orchestrate gene expression during somatic cell reprogramming and associated cell-state transitions. However, until recently, most studies have been performed in populations or subpopulations of reprogrammed cells, and only in recent years has it become possible to start interrogating these processes at the single-cell level [139,140,141,142,143]. Thus, an obvious challenge for the future is to fully understand how the gene regulatory mechanisms underlying somatic cell reprogramming fall into place at the single-cell level. In particular, future work will bring deeper insight into the temporal dynamics of TF interactions, epigenetic modifications, chromatin remodelling and DNA methylation in single cells.

In addition to reprogramming to pluripotency, the field has quickly extended to direct reprogramming into different cell identities. The core reprogramming complexes that are required to induce cell identity have also been identified in certain cases. For instance, direct reprogramming of fibroblasts to neurons requires the transcription factors ASCL1, BRN2 and Myt1l [144], while hemogenic reprogramming depends on GATA2, GFI1B and FOS expression [145]. Efficient reprogramming of somatic cells to hepatocytes [146] and cross-presenting dendritic cells (cDC1s) [147] has also been described. Further studies will elucidate which specific reprogramming machineries are required to induce additional cell identities. As in reprogramming to pluripotency, future work will likely be extended to single-cell studies in reprogramming to various other cell types.

What determines the functional interactions between regulatory factors and their target sequences in the genome is not completely understood. Given the recent boom in high-throughput (epi)genome-editing technologies [148,149,150], the functional interrogation of multiple putative reprogramming enhancers in specific loci and the dissection of their regulatory sequences via permutations, deletions and other modifications does not appear inconceivable. These approaches, in combination with novel imaging and live-cell imaging techniques [151,152,153], will constitute powerful tools to identify functional binding of key reprogramming TFs to better understand their binding dynamics. Altogether, this will also add to our current understanding of chromatin conformation in relation to gene expression and underpin which DNA features dictate regulatory output.

## Figures and Tables

**Figure 1 cells-11-01404-f001:**
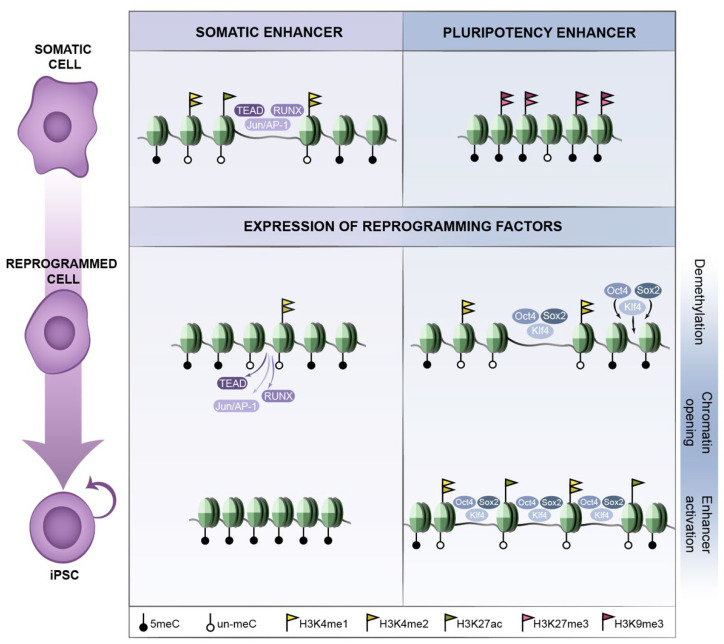
**Remodelling of epigenetic enhancer features during reprogramming to pluripotency.** Expression of reprogramming factors induces major epigenetic changes leading to repression of somatic enhancers and activation of the pluripotency regulatory landscape. The somatic enhancers (**left**), that in somatic cells are typically bound by active enhancer marks (i.e., H3K27ac, H3K4me1/2) and somatic TFs (e.g., TEAD, RUNX, Jun/AP-1), undergo epigenetic changes during the reprogramming process. This includes loss of active enhancer marks and eviction of somatic TFs, acquiring a closed methylated state in the resulting iPSCs. In contrast, pluripotent enhancers (**right**) are bound by repressive marks in somatic cells (i.e., H3K27me3, H3K9me3), and during the reprogramming process they gradually undergo demethylation, chromatin opening and eventually enhancer activation. The pluripotent active enhancers are decorated by active enhancer marks and bound by the OSK reprogramming factors.

**Figure 2 cells-11-01404-f002:**
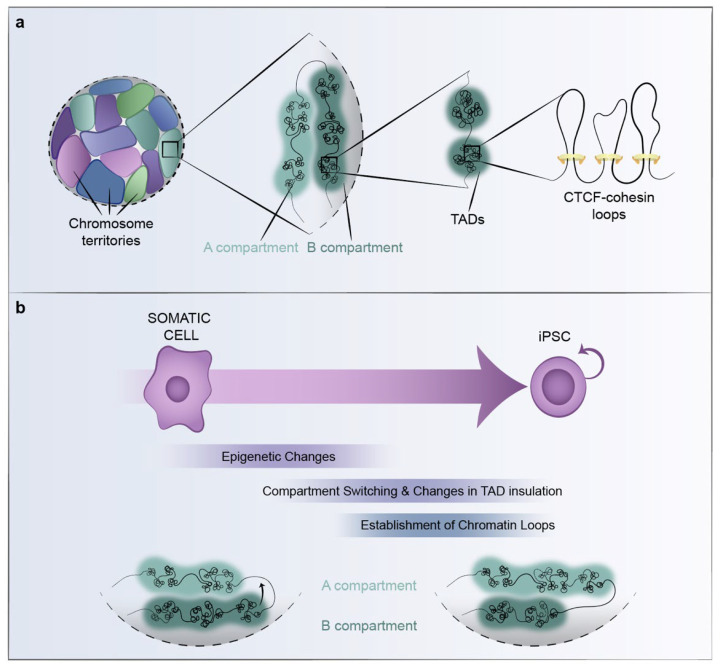
**3D chromatin organization and reprogramming.** (**a**) *The spatial organization of the genome at multiple scales:* Individual chromosomes occupy distinct chromosome territories, within which active and inactive chromatin segregate into A and B compartments, respectively. A-compartments localize towards the interior of the nucleus, while B-compartments are positioned towards the periphery and are enriched at the nuclear lamina. At smaller scales, chromatin is organized in TADs within which gene expression is controlled by functional dynamic interactions between promoters and enhancers. CTCF and cohesin play an architectural role in the 3D organization of the genome. (**b**) *Dynamics of genome organization during reprogramming:* Reprogramming to pluripotency involves a series of changes in the genome organization of the somatic cells into that of the iPSCs, including epigenetic changes, compartment switching, changes in TAD insulation and establishment of chromatin loops. The bottom panel represents compartment switching during reprogramming—here illustrated by a domain at the B-compartment in somatic cells that switches into the A-compartment in iPSCs.

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
