# Peer review of "Epigenetics, Enhancer Function and 3D Chromatin Organization in Reprogramming to Pluripotency"

_cells, 2022, doi:10.3390/cells11091404_

Round 1

Reviewer 1 Report

In the manuscript “Epigenetics, Enhancer Function and 3D Chromatin Organization in Reprogramming to Pluripotency”, the authors provided a review for the relations between epigenetics, 3D chromatin organization and remodeling of enhancer activity in somatic cell reprogramming process.

This is an interesting review related with regulatory mechanism of cellular reprogramming.

I only have one major comment. The authors mentioned the concept of gene regulatory networks. But they failed to discuss that changes in the regulatory networks are key to determine cell fates during embryonic development and reprogramming by changing the epigenetic landscape (J. Wang, et al., PNAS,108, 8257 (2011), Li et al. PLoS Comut. Biol. 9: e1003165 (2013)).

Reviewer 2 Report

In “Epigenetics, Enhancer Function and 3D Chromatin Organization in Reprogramming to Pluripotency”, the authors review the current knowledge on the processes that control the establishment and maintenance of pluripotency during somatic cell reprogramming. For this manuscript, there are some minor improvements needed to revise as detailed below.

1)    Please add some relevant references in the first paragraph of Introduction.

2)    In Lines 188-192, the authors clearly elucidate the features of active enhancers. Based on current knowledge, please indicate how to define inactive state of enhancers (silenced, primed or, poised) depending on combination of histone marks and other chromatin features?

3)  In Figure 2, the authors generally illustrate the epigenetic changes from somatic cell reprogramming to pluripotency. Please expand the specific changes of epigenetics in the figure.

4)  To make readers better understand the meaning of the figures, please add a more detailed description of the figures in the caption.

5)  The titles of Section 3 and 3.2 have mark mistakes. Please check and revise them in manuscript.

Reviewer 3 Report

This is a nice piece of review focusing on “Epigenetics, Enhancer Function and 3D Chromatin Organization in Reprogramming to Pluripotency” by Andreas Hörnblad and Silvia Remeseiro. They have nicely demonstrated the intricate molecular biology of reprogramming and how to meet pluripotency. Nevertheless, this reviewer is critical of two major points that need sincere attention before its possible acceptance. These points are,

  1. An updated report may be in a tabular form or with a timeline fashion, citing the journey to meet the successful induced pluripotency (iPSCs) generated upon somatic cell reprogramming, need to be incorporated in this review.
  2. List all the necessary transcription factors required other than OSKM to reprogram fibroblast to any other lineages that could be established with essential references in a tabular form for curious readers to follow up.

Reviewer 4 Report

An excellent and timely review of epigenetic reprogramming in somatic cells as they transition to pluripotency by ectopic expression of transcription factors/small molecules.

A minor point would be to include a brief comparison of important epigenetic marks on embryonic stem cells versus induced pluripotent stem cells

Author Response

Please notice that on April 8th, date when we were requested to revise this manuscript, this reviewer was listed as reviewer 3 and now has been renamed as reviewer 4.

Round 2

Reviewer 3 Report

None